# *Ecklonia stolonifera* Okamura Extract Suppresses Myocardial Infarction-Induced Left Ventricular Systolic Dysfunction by Inhibiting p300-HAT Activity

**DOI:** 10.3390/nu14030580

**Published:** 2022-01-28

**Authors:** Takahiro Katagiri, Yoichi Sunagawa, Tatsuya Maekawa, Masafumi Funamoto, Satoshi Shimizu, Kana Shimizu, Yasufumi Katanasaka, Maki Komiyama, Philip Hawke, Hideo Hara, Kiyoshi Mori, Koji Hasegawa, Tatsuya Morimoto

**Affiliations:** 1Division of Molecular Medicine, School of Pharmaceutical Sciences, University of Shizuoka, Shizuoka 422-8526, Japan; m17028@u-shizuoka-ken.ac.jp (T.K.); y.sunagawa@u-shizuoka-ken.ac.jp (Y.S.); m15100@u-shizuoka-ken.ac.jp (T.M.); funamoto@u-shizuoka-ken.ac.jp (M.F.); s18410@u-shizuoka-ken.ac.jp (S.S.); s18804@u-shizuoka-ken.ac.jp (K.S.); katana@u-shizuoka-ken.ac.jp (Y.K.); kiyoshimori2001@gmail.com (K.M.); koj@kuhp.kyoto-u.ac.jp (K.H.); 2Division of Translational Research, Clinical Research Institute, National Hospital Organization Kyoto Medical Center, Kyoto 612-8555, Japan; nikonikomakirin@yahoo.co.jp; 3Shizuoka General Hospital, Shizuoka 420-8527, Japan; 4Laboratory of Scientific English, School of Pharmaceutical Sciences, University of Shizuoka, Shizuoka 422-8526, Japan; hawke@u-shizuoka-ken.ac.jp; 5UNIAL Co., Ltd., Tokyo 173-0004, Japan; h-hara@unial.co.jp; 6Graduate School of Public Health, Shizuoka Graduate University of Public Health, Shizuoka 420-0881, Japan

**Keywords:** *Ecklonia stolonifera* Okamura, heart failure, myocardial infarction, p300, HAT activity

## Abstract

*Ecklonia stolonifera* Okamura extract (ESE) has been reported to have various bioactive effects, but its effects on cardiovascular disease have not yet been investigated. First, primary neonatal rat cultured cardiomyocytes were treated with ESE and stimulated with phenylephrine (PE) for 48 h. ESE (1000 µg/mL) significantly suppressed PE-induced cardiomyocyte hypertrophy, hypertrophy-related gene transcription, and the acetylation of histone H3K9. An in vitro p300-HAT assay indicated that ESE directly inhibited p300-HAT activity. Next, one week after myocardial infarction (MI) surgery, rats (left ventricular fractional shortening (LVFS) < 40%) were randomly assigned to three groups: vehicle (saline, *n* = 9), ESE (0.3 g/kg, *n* = 10), or ESE (1 g/kg, *n* = 10). Daily oral administration was carried out for 8 weeks. After treatment, LVFS was significantly higher in the ESE (1 g/kg) group than in the vehicle group. The ESE treatments also significantly suppressed MI-induced increases in myocardial cell diameter, perivascular fibrosis, hypertrophy- and fibrosis-related gene transcription, and the acetylation of histone H3K9. These results suggest that ESE suppressed both hypertrophic responses in cardiomyocytes and the development of heart failure in rats by inhibiting p300-HAT activity. Thus, this dietary extract is a potential novel therapeutic strategy for heart failure in humans.

## 1. Introduction

Heart failure (HF) is a clinical syndrome caused by a structural and/or functional cardiac abnormality, resulting in reduced cardiac output and elevated intracardiac pressure at rest or during stress [1]. The incidence of HF increases with age, and the number of patients with HF is increasing rapidly along with the aging of the population [2], so addressing this problem is of great social and clinical importance.

When the heart is stressed, the sympathetic nervous system and the renin-angiotensin system are activated, and these factors act on the heart to cause hypertrophy in individual cardiomyocytes [3,4]. These factors bind to receptors on the cell surface and, through intracellular signaling pathways, eventually reach the cardiomyocyte nucleus. In the nucleus, the histone acetyltransferase (HAT) activity of the transcriptional coactivator p300 is increased. This enhances the acetylation of the histone H3K9 and activates the transcription of hypertrophy-related genes [5,6,7].

Drugs that have been used to treat HF include beta-adrenergic receptor blockers, angiotensin II receptor blockers, and angiotensin-converting enzyme inhibitors, which act upstream in the signaling pathway that has been enhanced by stress [1,8,9,10,11]. However, even with the use of these drugs, the mortality rate of HF is still high, with approximately 50% of patients dying within 5 years of initial diagnosis [12,13]. There is a pressing need to develop new drugs for the treatment and prevention of HF.

Many types of algae have been traditionally used as foods in Japan, a country completely surrounded by ocean. As algae are rich in minerals and dietary fiber, and they are considered to be a healthy food. [14]. *Ecklonia stolonifera* Okamura (*E. stolonifera*), a perennial brown alga, is a member of the family Laminariaceae, belonging to the order Laminariales [15]. It grows mainly in the Sea of Japan [14]. *E. stolonifera* contains compounds such as phlorotannins [15], fucoxanthin [16], and alginic acid [17]. Phlorotannins, the main components of E. stolonifera, are a type of polyphenol. *E. stolonifera* contains the phlorotannins eckol, diekcol, phlorofucofuroeckol A, and eckstolonol [15]. Various bioactive effects of *E. stolonifera* have been reported, including antioxidant [18], anti-inflammatory [19], anti-diabetic [14,20], hepatoprotective [21], anti-obesity [22], and anti-tyrosinase effects [18].

While *E. stolonifera* has been reported to have various bioactive effects, its effects on cardiovascular disease have not yet been investigated. In this study, we investigate whether *E. stolonifera* extract (ESE) suppresses both hypertrophic responses in primary cultured neonatal rat cardiomyocytes and the development of HF in rats with myocardial infarction (MI).

## 2. Materials and Methods

### 2.1. Materials

ESE was provided by UNIAL Co., Ltd. (Tokyo, Japan) and stored at 4 °C. ESE was dissolved in distilled water, filtered, and then used in experiments.

### 2.2. Animal Experiments

Neonatal Sprague-Dawley (SD) rats were purchased from Japan SLC Inc. (Shizuoka, Japan). 7- to 8-week-old male SD rats were purchased from Japan CLEA Japan, Inc. (Tokyo, Japan). All animal experiments were performed according to the Guide for the Care and Use of Laboratory Animals published by the University of Shizuoka and the Kyoto Medical Center. This study was approved by the Ethics Committee of the University of Shizuoka (US176278 for primary cultured cardiomyocytes, US176279 for animal experiments) and the Kyoto Medical Center (KMC30-26-2 for animal experiments) in Japan.

### 2.3. Primary Cultured Neonatal Rat Cardiomyocytes

Primary cultured neonatal rat cardiomyocytes were isolated and seeded from 1- to 2-day-old SD rats as described previously [23,24]. Briefly, these cells were treated either with 100, 300, or 1000 μg/mL of ESE, or with saline as a vehicle control, for 2 h, and then stimulated with 30 μM phenylephrine (PE) for 48 h.

### 2.4. Immunofluorescence Staining

Immunofluorescence staining was performed as described previously [25]. Briefly, cardiomyocytes were fixed by 3.7% paraformaldehyde for 10 min and incubated with mouse anti-α-actinin antibody (Sigma-Aldrich, Saint Louis, MO, USA) for 1 h. After washing, these cells were incubated with Alexa Fluor 555-conjugated goat anti-mouse IgG secondary antibody (Thermo Fisher Scientific, Carlsbad, CA, USA) and Hoechst 33,258 (Dojindo, Kumamoto, Japan) for 1 h. After further washing, 300 α-actinin-positive cardiomyocyte surface areas were automatically measured by ArrayScan^TM^ (Thermo Fisher Scientific, Carlsbad, CA, USA).

### 2.5. Quantitative Reverse Transcription PCR

Quantitative reverse transcription PCR (RT-PCR) was performed as described previously [23,24]. In brief, the extraction of total RNA from cardiomyocytes and the left ventricles (LV) of rats was performed using TRIzol reagent (Invitrogen, Waltham, MA, USA). The synthesis of cDNA was performed using ReverTra Ace^®^ qPCR RT Master Mix (Toyobo, Osaka, Japan). Quantitative RT-PCR was performed with a LightCycler 96 Real-Time PCR System (Roche, Rotkreuz, Switzerland) using KOD SYBR qPCR Master Mix (Toyobo). The relative values of target genes were calculated using the ΔΔCT method and normalized using rat HPRT1 genes as an internal control. Primers used for quantitative RT-PCR are shown in Appendix A.

### 2.6. Western Blotting

Histone fractions were isolated from primary cultured neonatal rats and SD rats as described previously [23,26]. The fractions were subjected to SDS-PAGE followed by Western blotting. Anti-acetyl-histone H3K9 (#9649, Cell Signaling Technology, Danvers, MA, USA) and anti-histone H3 (#4499, Cell Signaling Technology, USA) antibodies were used for Western blotting. Chemiluminescent signals were visualized using an Amersham Imager 680 (GE Healthcare Life Science, São Paulo, SP, Brazil) and quantified with Image J (1.52v).

### 2.7. In Vitro p300-HAT Assay

In Vitro p300-HAT assays were performed using purified p300-HAT recombinant domain (residues 1284–1674) as described previously [27]. Then, 5 μg of histone from calf thymus (Worthington, Columbus, OH, USA) was incubated in HAT buffer with a p300-HAT recombinant domain in the presence or absence of ESE at room temperature for 30 min, followed by the addition of 1 μL acetyl-CoA for 1 h in a 20 μL reaction mixture. To the mixture was added to 6.6 μL 4× SDS-PAGE sample buffer, then the mixture was boiled for 10 min, and then subjected to Western blotting using anti-histone H3K9 and anti-histone H3 antibodies. The 50% inhibitory concentration (IC_50_) was calculated from the concentration–response curve.

### 2.8. MI Surgery

Eight-week-old male SD rats (250–300 g) were anaesthetized with 10% chloral hydrate, and their limbs were fixed to a plate. MI was created in these rats by ligating the proximal left anterior descending (LAD) coronary artery through a left thoracotomy, as described previously [28]. The same surgical procedure was performed in sham-surgery rats in which the LAD coronary artery was not ligated. At 1 week after surgery, the cardiac function of all surviving rats was noninvasively evaluated by echocardiography using a 10–12 MHz phased-array transducer (model 21380A) with an HP Sonos 5500 imaging system (Agilent Technologies, Santa Clara, CA, USA), as described previously [29]. The left ventricular internal diameter at end-diastole (LVIDd), the posterior wall diameter (PWD), and the LV fractional shortening (LVFS) were measured with M-mode tracings from the short-axis view of the LV at the papillary muscle level. All measurements were performed in a blinded fashion according to the guidelines of the American Society for Echocardiology and averaged over three consecutive cardiac cycles. Pulse rate (PR), diastolic blood pressure (DBP), and systolic blood pressure (SBP) were measured in all surviving rats with the tail-cuff method (BP-98A, Softron, Tokyo, Japan).

### 2.9. Treatment

At 1 week after MI surgery, MI rats with FS <40% were randomly assigned to four groups: Group I (*n* = 6) comprised sham-surgery rats with vehicle treatment (1% gum arabic) as a control. Groups II–IV comprised MI rats with vehicle (*n* = 9), ESE (0.3 g/kg/day, *n* = 10), and ESE (1 g/kg/day, *n* = 10). Daily oral administration using a sonde was carried out for 8 weeks. ESE was suspended in 1% gum arabic solution. We administrated about 250 mg ESE in 2 mL 1% gum arabic solution to rats with a body weight (BW) of 250 g. Although ESE was not completely dissolved in the 1% gum arabic solution, administrating it to rats using a sonde proceeded smoothly.

### 2.10. Histological Analysis

The rats were euthanized, and their hearts were isolated and cut into two transverse slices at the mid-level of the papillary muscles. The heart tissues were fixed in 10% formalin, embedded in paraffin, sliced into 5 μm thick sections, and stained with hematoxylin eosin (HE) and picrosirius red (PSR) as described previously [23]. The sections were deparaffinized and incubated with PSR (1.2% picric acid (FUJIFILM Wako Pure Chemicals Corporation, Osaka, Japan)), 0.1% Direct Red 80 Plus (Sigma-Aldrich, Saint Louis, MO, USA), and 0.1% Fast Green FCF (Sigma-Aldrich, Saint Louis, MO, USA)) for 60 min while being protected from light. Quantitative assessments of cross-sectional myocardial cell diameter and perivascular fibrosis area were carried out as previously described [23]. HE-stained and PSR-stained sections were photographed using an Eclipse 80i microscope (Nikon, Tokyo, Japan). At least 50 myocardial cells with a nucleus were evaluated, and their diameter was measured using ImageJ software. Areas of perivascular fibrosis were measured using ImageJ software, and the resulting value divided by the area of the intramyocardial coronary artery was regarded as the relative vascularized fibrosis area. The scale of the measured intramyocardial coronary artery was more than 50 μm in each rat.

### 2.11. Statistical Analysis

Results are presented as the mean ± SE. Statistical comparisons were performed using one-way ANOVA with the Tukey–Kramer test for post hoc multiple comparison (Stat View 5.0 software, SAS Institute Inc., Cary, NC, USA). *p* < 0.05 was considered statistically significant.

## 3. Results

### 3.1. ESE Suppressed PE-Induced Hypertrophic Responses in Cardiomyocytes

To investigate whether ESE suppresses PE-induced cardiomyocyte hypertrophy, primary cultured cardiomyocytes were treated with 100, 300, or 1000 μg/mL of ESE for 2 h and then stimulated with 30 µM PE for 48 h. As shown in Figure 1A,B, ESE dose-dependently inhibited PE-induced cardiomyocyte hypertrophy. The results of quantitative RT-PCR revealed that ESE significantly suppressed PE-induced increases in the transcription of ANF and BNP (Figure 1C,D). Western blotting showed that ESE also suppressed a PE-induced increase in acetylation of histone H3K9 (Figure 1E,F). These results indicate that ESE suppressed PE-induced hypertrophic responses in cardiomyocytes at least in part by inhibiting p300-HAT activity.

### 3.2. ESE Inhibited p300-HAT Activity In Vitro

To determine whether ESE directly inhibits p300-HAT activity, we performed an in vitro p300-HAT assay using a recombinant p300-HAT domain (1284–1674) in addition to ESE. The results of this assay, shown in (Figure 2A), indicated that the acetylation of histone H3K9 was significantly inhibited by ESE. The IC_50_ value of ESE was calculated from sigmoid dose–response curves with variable slopes (Figure 2B), resulting in an estimated value of 505 μg/mL. These results suggest that ESE directly inhibited p300-HAT activity.

### 3.3. ESE Improved MI-Induced Systolic Disfunction in Rats

A rat MI model was used to determine whether ESE suppresses the development of heart failure in vivo. One week after MI surgery, surviving rats with moderate MI (FS < 40%) were subjected to echocardiographic and hemodynamic parameter measurements and then divided into three groups: ESE (0.3 g/kg/day), ESE (1 g/kg/day), and vehicle (1% gum arabic) as a control (Table 1). MI surgery decreased LVFS and enlarged LVIDd compared to sham surgery. There were no differences in LVIDd, PWD, LVFS, SBP, DBP, HR, or BW among the three MI groups before administration. Daily oral administration was carried out for 8 weeks. Representative echocardiography images are shown in Figure 3A, and echocardiographic and hemodynamic parameters in Table 2. At 8 weeks after treatment, the vehicle group had significantly decreased LVFS and increased PWD. These changes were improved by ESE treatment. The vehicle group also showed significant increases in calculated LV mass index and heart weight to tibia length (HW/TL) ratio, and these increases were also reduced by ESE treatment (Figure 3B,C). Representative images of PSR-stained sections of LV myocardium are shown in Figure 3D. There was no difference among the three groups in infarct size (15.2%–16.1%, Table 2). These results indicate that ESE treatment significantly suppressed cardiac hypertrophy and preserved systolic function in rats after MI.

### 3.4. ESE Suppressed MI-Induced Cardiac Hypertrophy in Rats

To investigate whether ESE suppresses MI-induced cardiac hypertrophy in rats, HE staining was performed using sections of LV myocardium. Representative images of HE-stained cardiomyocytes are shown in Figure 4A. The results of cross-sectional myocardial cell diameter measurement showed that ESE treatment significantly but not completely suppressed MI-induced cardiomyocyte hypertrophy (Figure 4B). Next, to investigate whether ESE suppresses MI-induced hypertrophy-related gene transcription in rats, quantitative RT-PCR was performed. The results indicated that ESE treatment significantly suppressed the transcription of the hypertrophy-related genes ANF and BNP (Figure 4C,D). These findings indicate that ESE suppressed MI-induced cardiac hypertrophic responses in rats.

### 3.5. ESE Suppressed MI-Induced Cardiac Fibrosis in Rats

To investigate whether ESE suppresses MI-induced cardiac fibrosis in rats, PSR staining was performed using sections of LV myocardium. Representative images of PSR-stained perivascular fibrosis area are shown in Figure 5A. Fibrosis area measurements showed that ESE treatment significantly suppressed MI-induced perivascular fibrosis (Figure 5B). Next, to investigate whether ESE suppresses MI-induced fibrotic gene transcription in rats, quantitative RT-PCR was performed. The results indicated that ESE treatment significantly suppressed an MI-induced increase in the transcription of the fibrosis-related genes α-SMA and collagen 1A1 (Figure 5C,D). These results suggest that ESE suppressed MI-induced cardiac fibrosis in rats.

### 3.6. ESE Suppressed MI-Induced Acetylation of Histone H3K9 in Rats

To investigate whether ESE suppresses the MI-induced acetylation of histone H3K9 in rats, Western blotting was performed using histone fractions from the rat hearts. The results indicated that ESE treatment significantly suppressed the MI-induced acetylation of histone H3K9 (Figure 6A,B).

## 4. Discussion

This study found that ESE suppressed both PE-induced cardiomyocyte hypertrophic response in primary cultured cardiomyocytes as well as cardiac hypertrophy and left ventricular systolic dysfunction in MI rats. This finding is in line with our previous reports that natural compounds such as curcumin and metformin inhibit cardiomyocyte hypertrophic response and the development of heart failure.

In the present study, the findings of our in vitro p300-HAT, cultured cardiomyocyte, and MI rat model experiments suggest that ESE inhibits p300-HAT activity. We previously reported that curcumin and metformin specifically inhibit p300 HAT activity [7,25]. In that study, an in vitro p300-HAT assay showed that the IC_50_ of curcumin was 9.4 µM (3.4 µg/mL) [30]. In the present study, the in vitro p300-HAT assay showed that ESE inhibited the HAT activity of p300 in a concentration-dependent manner, with an IC_50_ of 505 µg/mL.

In addition, our previous studies found that curcumin inhibits the PE- and MI-induced acetylation of histone H3K9, which in turn inhibits the transcriptional activity of hypertrophy-related genes, cardiomyocyte hypertrophy, and the progression of heart failure [23]. In the present study, ESE also inhibited the PE- and MI-induced acetylation of histone H3K9 and suppressed cardiomyocyte hypertrophy and left ventricular systolic dysfunction, suggesting that ESE, like curcumin, may be a novel preventive and therapeutic agent for heart failure. However, as the IC_50_ of ESE for the inhibition of p300-HAT activity is much higher than that of curcumin, ESE requires administration in larger doses than the other compounds. This is because ESE is an extract containing a variety of compounds. A dose of 1 g/kg/day inhibited the progression of heart failure in MI rats; however, using this dose in clinical practice would require a 60 kg adult to consume 60 g of ESE per day, a dose that is far too high for practical use. Therefore, further research is needed to identify and extract the active components of ESE that inhibit p300-HAT activity and to improve the formulation of ESE in order to reduce the required dose.

Inflammation is known to be involved in the progression of heart failure. The release of inflammatory cytokines such as TNF-α, IL-1β, IL-6, and IL-8, and the production of inducible nitric oxide synthase (iNOS) and cyclooxygenase-2 (COX-2), can lead to cardiac hypertrophy and impaired cardiac function [31,32,33,34]. Oxidative stress is also involved in the progression of heart failure, with excessive production of reactive oxygen species leading to cardiomyocyte hypertrophy and mitochondrial dysfunction [35,36,37]. E. stolonifera is rich in phlorotannins, which have been reported to have antioxidant effects that inhibit the production of reactive oxygen species [18,38]. Among the phlorotannins, florofucofuroecol-A has been shown to have not only antioxidant effects, but also anti-inflammatory effects via the suppression of iNOS and COX-2 expression [19]. Fucoxanthin, a carotenoid, has been reported to have both anti-inflammatory effects via the suppression of iNOS expression and antioxidant effects via the inhibition of reactive oxygen species generation [39,40]. These findings suggest that, in addition to inhibiting p300-HAT activity, ESE may inhibit the progression of heart failure by exerting anti-inflammatory and antioxidant effects.

In this study, ESE inhibited perivascular fibrosis and the transcription of fibrosis-related genes in MI rats, suggesting that it inhibits fibrotic response after MI. Previous reports have shown that myocardial infarction causes reparative fibrosis, in which cardiomyocytes in the infarcted area become necrotic and are eventually replaced by fibroblasts and collagen fibers. This increase in the extracellular matrix in response to fibrosis causes a decrease in the diastolic and contractile capacities of the heart, thereby contributing to the transition to heart failure [41,42,43]. On the other hand, chronic stress-induced cardiac hypertrophy continuously activates fibroblasts, resulting in reactive fibrosis of the interstitium without cardiomyocyte shedding [44,45]. ESE may suppress this reactive fibrosis by inhibiting cardiomyocyte hypertrophy. In addition, inflammatory factors that are activated during the progression of heart failure are also involved in fibrosis [46]. This suggests that ESE may suppress fibrosis due to its suppression of inflammatory factors activated by stress due to myocardial infarction. It has also been reported that fucoxanthin inhibits a fibrotic response induced by TGF-β1 stimulation in nasal polyp-derived fibroblasts [47], suggesting that the anti-fibrotic effect of fucoxanthin may have contributed to the fibrotic response after MI in the present study. Further studies using cultured cardiac fibroblasts are needed to clarify whether or not ESE has an anti-fibrotic effect.

This study did not find any adverse effects of ESE. Daily oral administration of l g/kg of ESE for 8 weeks to MI rats did not cause any adverse effects or any difference in liver or kidney weight compared to the control. In addition, a repeated-dose toxicity study and a micronucleus study of ESE at a dose of 2 g/kg confirmed its safety. Furthermore, we believe that the hurdle for the clinical application of ESE is low due to the fact that it is used as a food.

In conclusion, this study found that ESE suppressed both cardiomyocyte hypertrophic response and the progression of heart failure after myocardial infarction in a rat model of MI. Further detailed studies may lead to the use of ESE as a novel preventive and therapeutic agent for heart failure.

## Figures and Tables

**Figure 1 nutrients-14-00580-f001:**
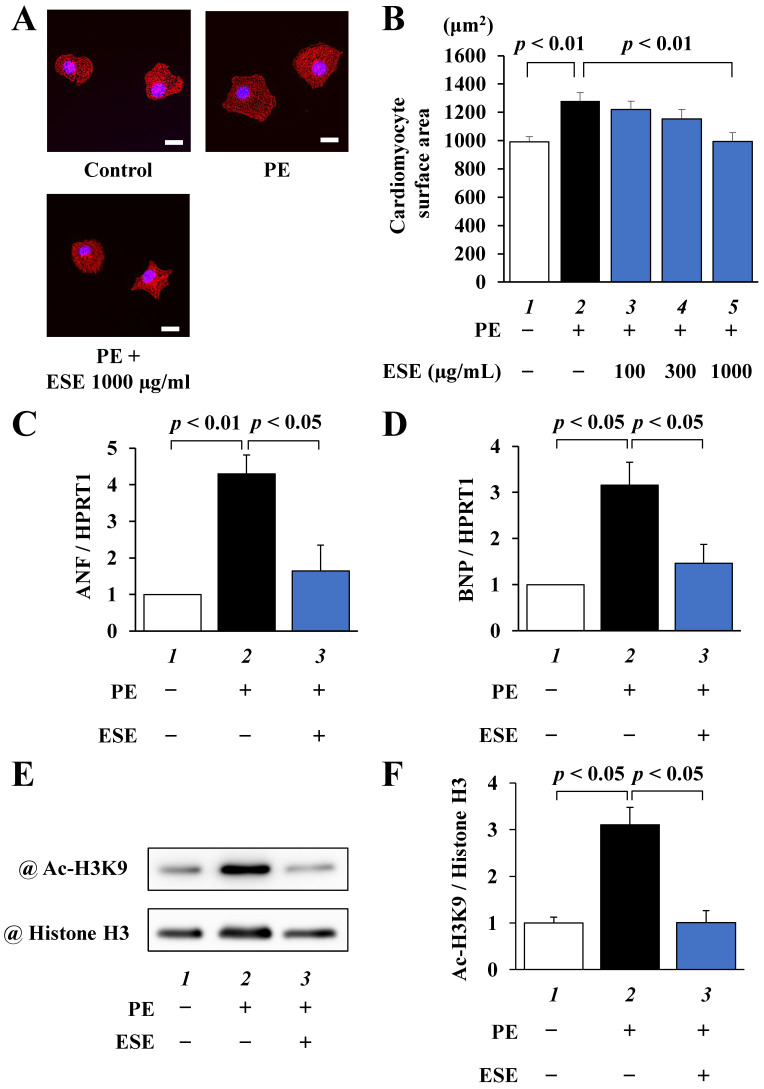
PE-induced hypertrophic responses were suppressed by ESE treatment in cardiomyocytes. Primary cultured cardiomyocytes were treated with 100, 300, or 1000 μg/mL ESE and stimulated with 30 μM PE. (**A**) Immunofluorescence staining was performed using anti-α-actinin antibodies and Alexa Fluor 555-labeled anti-mouse IgG. Scale bar: 20 μm. (**B**) Cardiomyocyte surface area measurements. All data are presented as the mean ± SEM of three individual experiments. (**C**,**D**) Total RNA was extracted from the cells, and quantitative RT-PCR was performed for (**C**) ANF, (**D**) BNP, and HPRT1. Data are presented as the mean ± SEM of three individual experiments. (**E**) Histone fractions isolated from the cells were subjected to Western blotting using anti-acetyl-histone H3K9 antibodies and anti-histone H3 antibodies. (**F**) Quantified levels of acetylated histone H3K9 and total histone H3. Data are presented as the mean ± SEM of three individual experiments.

**Figure 2 nutrients-14-00580-f002:**
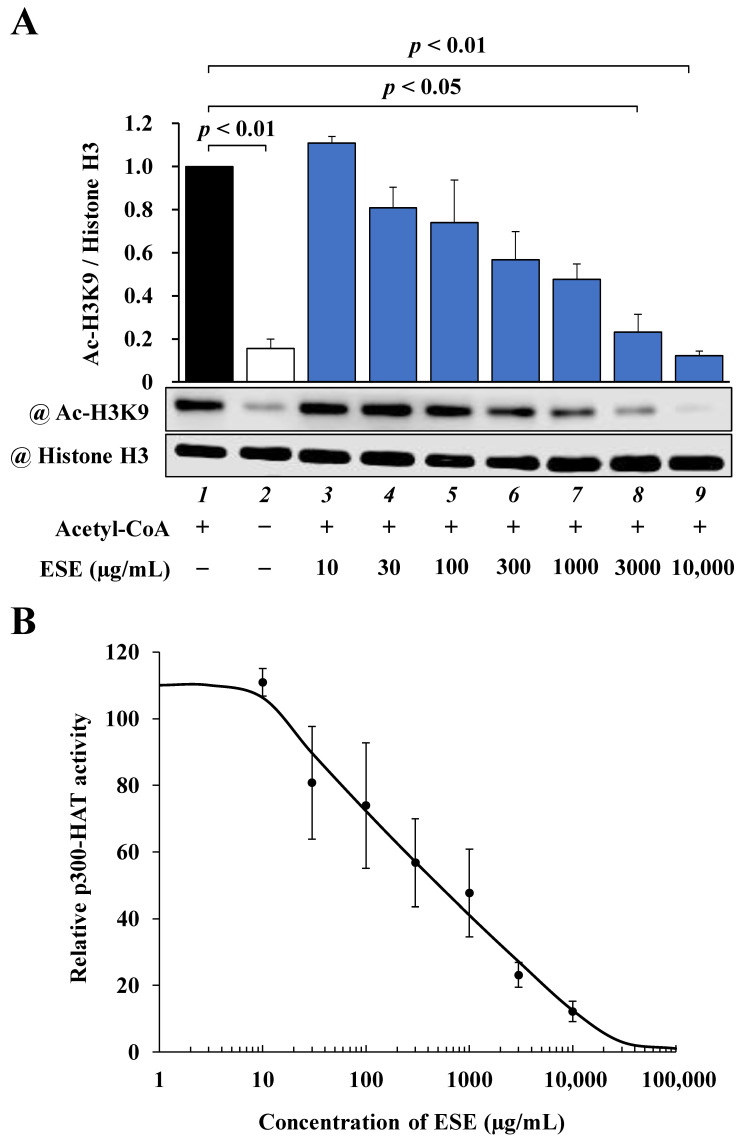
The effect of ESE on p300-HAT activity in vitro. (**A**) Samples were subjected to Western blotting with anti-acetyl-histone H3K9 antibodies and anti-histone H3 antibodies. Shown are quantified levels of acetylated histone H3K9 and total histone H3. Data are presented as the mean ± SEM of three individual experiments. (**B**) Concentration–response curve obtained by plotting acetyl-histone H3K9/histone H3 vs. common logarithm (concentrations). The IC50 value of ESE was 505 μg/mL. Quantified values of ESE are presented as the mean ± SEM of three individual experiments.

**Figure 3 nutrients-14-00580-f003:**
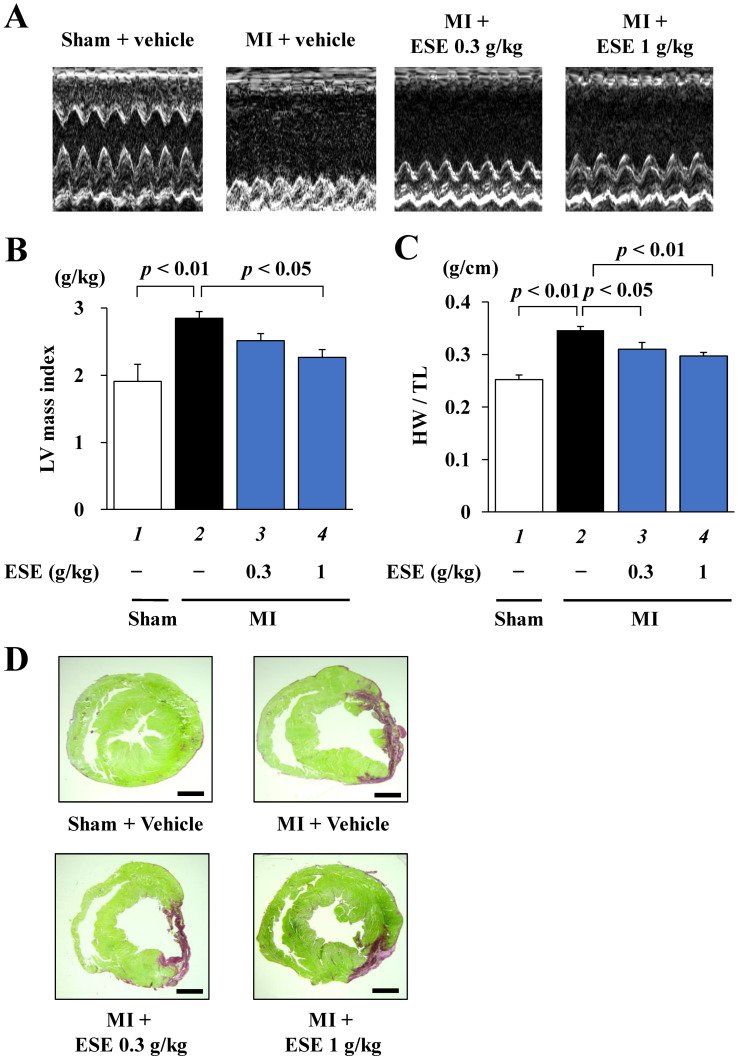
MI-induced systolic dysfunction was improved by ESE treatment in vivo. Cardiac function was measured by echocardiography at 8 weeks after treatment. (**A**) Representative images of echocardiography. (**B**) MI-induced increase in LV mass index was significantly suppressed by 1 g/kg of ESE. Data are presented as the mean ± SEM. (**C**) MI-induced increase in HW/TL was significantly suppressed by 1 g/kg of ESE. (**D**) Representative photographs of PSR-stained sections of LV myocardium from sham and MI rats. Scale bar: 2 mm.

**Figure 4 nutrients-14-00580-f004:**
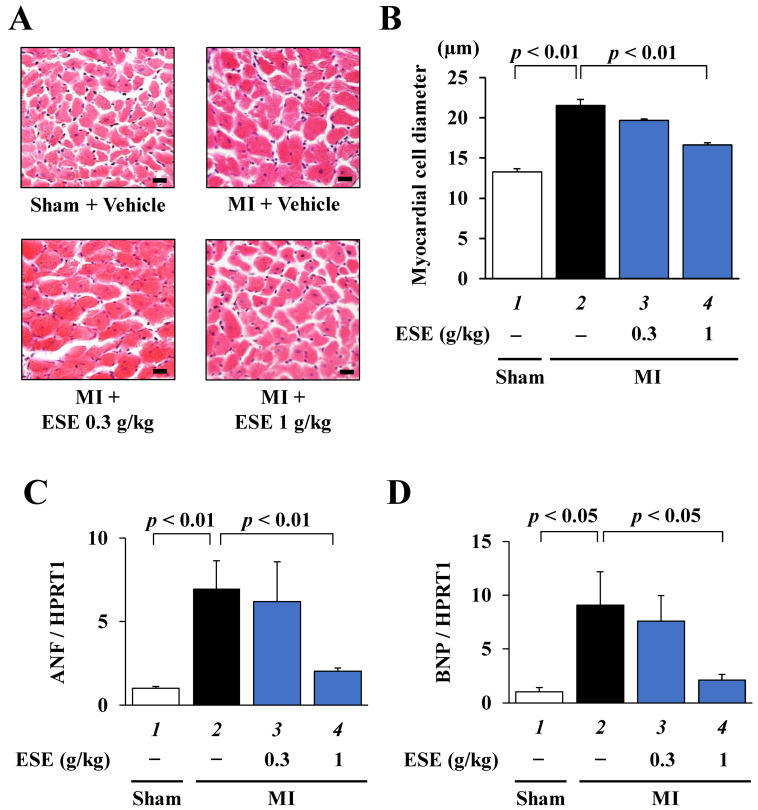
Cardiac hypertrophy was suppressed by ESE treatment in rats with MI. (**A**) Representative images of HE-stained sections of cardiomyocytes from sham and MI rats. Magnification: ×400. Scale bar: 20 μm. (**B**) Myocardial cell diameter was measured for at least 50 cells in each rat. Data are presented as the mean ± SEM. (**C**,**D**) Quantitative RT-PCR data for (**C**) ANF, (**D**) BNP, and HPRT1. Data are presented as the mean ± SEM.

**Figure 5 nutrients-14-00580-f005:**
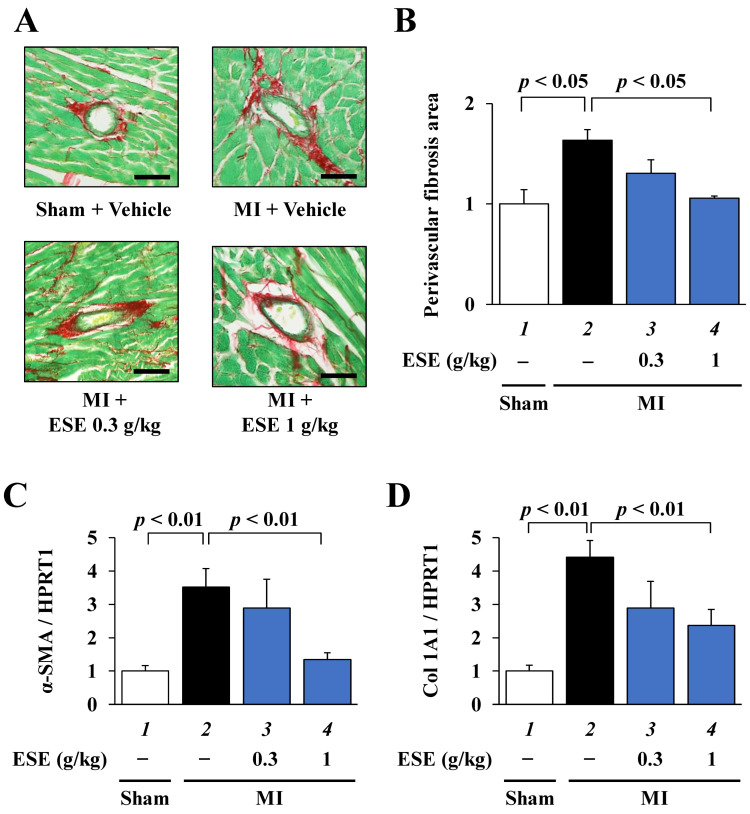
Cardiac perivascular fibrosis was suppressed by ESE treatment in rats with MI. (**A**) Representative images of PSR-stained perivascular fibrosis area of LV myocardium from sham and MI rats. Magnification: ×400. Scale bar: 50 μm. (**B**) Areas of perivascular fibrosis in the left ventricle were measured for at least three intramyocardial coronary arteries with a lumen size > 50μm. Data are presented as the mean ± SEM. (**C**,**D**) Quantitative RT-PCR data for (**C**) α-SMA, (**D**) Col 1A1, and HPRT1. Data are presented as the mean ± SEM.

**Figure 6 nutrients-14-00580-f006:**
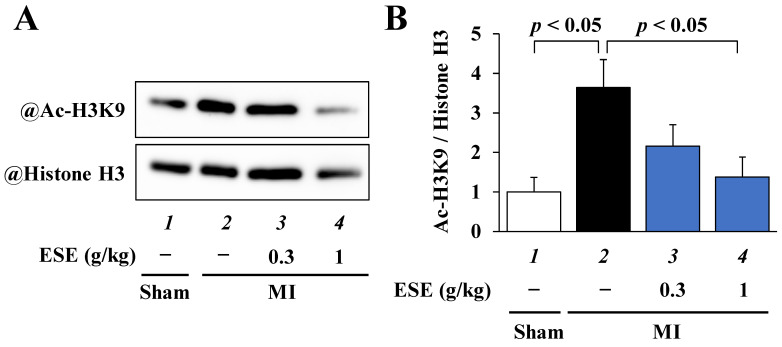
The acetylation of histone H3K9 was suppressed by ESE treatment in LV of rats with MI. (**A**) Histone fractions from the rat hearts were subjected to Western blotting to assess acetylated histone H3K9 and total histone H3 levels. (**B**) Quantified levels of acetylated histone H3K9 and total histone H3. Data are presented as the mean ± SEM.

**Table 1 nutrients-14-00580-t001:** Echocardiographic and hemodynamic parameters at 1 week after MI surgery.

	Sham	MI		
	Vehicle	Vehicle	ESE 0.3 g/kg	ESE 1 g/kg
LIVIDd (mm)	6.11 ± 0.21	8.26 ± 0.25 **	7.90 ± 0.24 **	7.75 ± 0.17 **
PWD (mm)	2.80 ± 0.10	2.78 ± 0.06	2.75 ± 0.11	2.82 ± 0.09
LVFS (%)	63.3 ± 3.9	25.3 ± 1.6 **	26.1 ± 1.1 **	25.9 ± 1.1 **
SBP (mmHg)	96 ± 6	99 ± 3	97 ± 4	98 ± 3
DBP (mmHg)	78 ± 2	74 ± 4	62 ± 6	72 ± 3
HR (bpm)	458 ± 15	445 ± 14	451 ± 10	466 ± 12
BW (g)	368 ± 11	372 ± 5	368 ± 8	347 ± 5

** *p* < 0.01 versus sham + vehicle. LVIDd, left ventricular internal diameter end-diastole; PWD, posterior wall diameter; FS, fractional shortening; SBP, systolic blood pressure; DBP, diastolic BP; HR, heart rate; BW, body weight.

**Table 2 nutrients-14-00580-t002:** Echocardiographic and hemodynamic parameters at 9 weeks after MI surgery.

	Sham	MI		
	Vehicle	Vehicle	ESE 0.3 g/kg	ESE 1 g/kg
LIVIDd (mm)	6.62 ± 0.12	9.64 ± 0.40 **	8.96 ± 0.38 **	8.77 ± 0.24 **
PWD (mm)	2.32 ± 0.13	3.18 ± 0.07 **	2.90 ± 0.11 *	2.56 ± 0.09 ##
LVFS (%)	58.2 ± 1.8	16.6 ± 1.3 **	18.1 ± 1.8 **	23.3 ± 0.7 **##
SBP (mmHg)	102 ± 4	104 ± 3	107 ± 3	108 ± 3
DBP (mmHg)	84 ± 16	75 ± 3	75 ± 5	87 ± 2
HR (bpm)	449 ± 14	460 ± 10	442 ± 10	431 ± 13
BW (g)	628 ± 15	614 ± 13	581 ± 11	562 ± 9 *##
Infarct size (%)	-	16.1 ± 1.3	15.5 ± 1.1	15.2 ± 3.6

* *p* < 0.05, ** *p* < 0.01 versus sham + vehicle, ## *p* < 0.01 versus MI + vehicle.

## Data Availability

The data in this study are available on request from the corresponding authors.

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
