# Peer review of "Ecklonia stolonifera Okamura Extract Suppresses Myocardial Infarction-Induced Left Ventricular Systolic Dysfunction by Inhibiting p300-HAT Activity"

_nutrients, 2022, doi:10.3390/nu14030580_

Round 1

Reviewer 1 Report

In this paper, the authors tested the effects of Ecklonia stolonifera Okamura extract in preventing hypertrophy of cultured rat cardiomyocytes and the development of HF in rats with myocardial infarction.

The experiments are adequately described and the results are clearly presented. I congratulate the authors for their work which demonstrated that Ecklonia stolonifera extract prevents cardiac remodeling in a cellular and animal model. It is thus supported the need for in-human studies.

Unfortunately, the files provided did not have any tables and figures for review. Maybe there was a problem with the file build-up.

Regarding the text I have just two minor observations:

  1. the work has been previously presented in abstract form (https://doi.org/10.1161/res.129.suppl_1.P381Circulation Research. 2021;129:AP381) and the abstract of the manuscript is very similar to the abstract for the poster. If considered appropriate, the authors may rephrase the abstract.
  2. in the abstract - line 29, the correct name for the parameter of left ventricular function is left ventricular fractional shortening.

Author Response

Review #1:

Comment:

              In this paper, the authors tested the effects of Ecklonia stolonifera Okamura extract in preventing hypertrophy of cultured rat cardiomyocytes and the development of HF in rats with myocardial infarction.

              The experiments are adequately described and the results are clearly presented. I congratulate the authors for their work which demonstrated that Ecklonia stolonifera extract prevents cardiac remodeling in a cellular and animal model. It is thus supported the need for in-human studies.

              Unfortunately, the files provided did not have any tables and figures for review. Maybe there was a problem with the file build-up.

Response:

              We really appreciate the reviewer for the time and expertise spent in reviewing our manuscript. We are truly sorry about out submitted files which lack any tables and figures. We resubmitted our revised manuscript including all of tables, figures, and figure legends in a ZIP file.

Comment:

              Regarding the text I have just two minor observations:

  1. the work has been previously presented in abstract form (https://doi.org/10.1161/res.129.suppl_1.P381Circulation Research. 2021;129:AP381) and the abstract of the manuscript is very similar to the abstract for the poster. If considered appropriate, the authors may rephrase the abstract.

Response:

              Thank you for your comment. As you indicated, the abstract was a proceeding. We have performed a poster presentation in American Heart Association Scientific Sessions 2021 and this abstract was published at Circulation Research as a proceeding. The authors of the presentation are the same to our submitted manuscript.

Comment:

  1. in the abstract - line 29, the correct name for the parameter of left ventricular function is left ventricular fractional shortening.

Response:

              Thank you for your point. We corrected it in the revised manuscript.

Reviewer 2 Report

Katagiri et al. investigated to administration of Ecklonia stolonifera Okamura extract (ESE) in a rat model of myocardial infarction. The authors found that ESE suppressed MI-induced increases in myocardial cell diameter, perivascular fibrosis, hypertrophy and fibrosis

Are US176278, US176279 and KMC30-26-2 ethics vote numbers? If yes, please state more clearly.

Was a special handling required to administer ESE orally to rats? Was it possible to safely administer the complete dose, or were there problems sometimes (vomiting?)

The authors present a highy interesting paper, the methodology seems very sound and extensive experimental has been done.

However, I could not find the referenced figures in the document, also figure legends were missing. Have the authors forgotten to submit the figures?

The paper is of interest, but a concise review can only be performed when all necessary figures and data are available. I recommend to resubmit the manuscript.

Author Response

Reviewer #2:

Comment:

              Katagiri et al. investigated to administration of Ecklonia stolonifera Okamura extract (ESE) in a rat model of myocardial infarction. The authors found that ESE suppressed MI-induced increases in myocardial cell diameter, perivascular fibrosis, hypertrophy and fibrosis

Response:

              We are thankful to the reviewer for the expert review of our manuscript and kind comments. According to your suggestion, we modified the revised manuscript. 

Comment:

              Are US176278, US176279 and KMC30-26-2 ethics vote numbers? If yes, please state more clearly.

Response:

              Thank you for your point. These are the approval number for this study. We added the details of ethics vote numbers in the revised manuscript.

Line 77-84,

“2.2. Animal experiments

Neonatal Sprague-Dawley (SD) rats were purchased from Japan SLC Inc. 7- to 8-week-old male SD rats were purchased from Japan Clea Co., Ltd. (Japan). All animal experiments were performed according to the Guide for the Care and Use of Laboratory Animals published by the University of Shizuoka and the Kyoto medical center. This study approved by the Ethics Committee of the University of Shizuoka (US176278 for primary cultured cardiomyocytes, US176279 for an animal experiment) and the Kyoto Medical Center (KMC30-26-2 for an animal experiment) in Japan.”

Line 358-362,

“Institutional Review Board Statement: All animal experiments were approved and carried out according to the Guide for the Care and Use of Laboratory Animals by the Institute of Laboratory Animal Care and Use Committee, University of Shizuoka (US176278 for primary cultured cardiomyocytes, US176279 for an animal experiment) and Kyoto Medical Center (KMC30-26-2 for an animal experiment) in Japan.”

Comment:

Was a special handling required to administer ESE orally to rats? Was it possible to safely administer the complete dose, or were there problems sometimes (vomiting?)

Response:

              Thank you for your valuable comment. ESE was suspended in 1% gum arabic solution. We administrated about Although ESE was not completely dissolved in 1% gum arabic solution, there were no problems to administer to rats using sonde.

We added them in the revised manuscript.

Line 144-152,

“2.9. Treatment

At 1 week after MI surgery, MI rats with FS < 40% were randomly assigned to four groups: Group I (n=6) comprised sham-surgery rats with vehicle treatment (1% gum arabic) as a control. Groups II–IV comprised MI rats with vehicle (n=9), ESE (0.3 g/kg/day, n=10), and ESE (1 g/kg/day, n=10). Daily oral administration using a sonde was carried out for 8 weeks. ESE was suspended in 1% gum arabic solution. We administrated about 250 mg ESE in 2 mL 1% gum arabic solution to 250 g (rat BW). Although ESE was not completely dissolved in 1% gum arabic solution, there were no problems to administer to rats using sonde.”

Comment:

The authors present a highy interesting paper, the methodology seems very sound and extensive experimental has been done.

However, I could not find the referenced figures in the document, also figure legends were missing. Have the authors forgotten to submit the figures?

The paper is of interest, but a concise review can only be performed when all necessary figures and data are available. I recommend to resubmit the manuscript.

Response:

              We are truly sorry about out submitted files which lacks any tables and figures. We resubmitted our revised manuscript including all of tables, figures, and figure legends in a ZIP file.

Round 2

Reviewer 1 Report

I thank the authors for revising the manuscript and providing tables and figures.

There are some minor issues that need to be addressed:

  1. Introduction line 28 should read: "left ventricular fractional shortening"
  2. Table 1 should be inserted in results section 3.3.
  3. Some references have inverted surname/first names. Maybe recheck all.

For example refs 1, 12, 13, 31, 33, 37, 44, 45, 46

  1. Reference 1 should be: Ponikowski P, Voors AA, Anker SD, Bueno H, Cleland JGF, Coats AJS, Falk V, González-Juanatey JR, Harjola VP, Jankowska EA, Jessup M, Linde C, Nihoyannopoulos P, Parissis JT, Pieske B, Riley JP, Rosano GMC, Ruilope LM, Ruschitzka F, Rutten FH, van der Meer P; ESC Scientific Document Group. 2016 ESC Guidelines for the diagnosis and treatment of acute and chronic heart failure: The Task Force for the diagnosis and treatment of acute and chronic heart failure of the European Society of Cardiology (ESC)Developed with the special contribution of the Heart Failure Association (HFA) of the ESC. Eur Heart J. 2016 Jul 14;37(27):2129-2200. doi: 10.1093/eurheartj/ehw128. Epub 2016 May 20. Erratum in: Eur Heart J. 2016 Dec 30;: PMID: 27206819.
  2. Ref 12: Riehle C, Bauersachs J. Small animal models of heart failure. Cardiovasc Res. 2019 Nov 1;115(13):1838-1849. doi: 10.1093/cvr/cvz161. PMID: 31243437; PMCID: PMC6803815.

so on...

Author Response

Reviewer 1

Comment:

I thank the authors for revising the manuscript and providing tables and figures.

There are some minor issues that need to be addressed:

Response:

Thank you very much for providing important comment. We are grateful for the time and energy you expended on our behalf.

Introduction line 28 should read: "left ventricular fractional shortening"

Response:

Thank you for your point. We corrected it in the revised manuscript.

Comment:

Table 1 should be inserted in results section 3.3.

Response:

Thank you for your comment. We inserted Table 1 after the results section 3.3 in the revised manuscript.

Comment:

Some references have inverted surname/first names. Maybe recheck all.

For example refs 1, 12, 13, 31, 33, 37, 44, 45, 46

Reference 1 should be: Ponikowski P, Voors AA, Anker SD, Bueno H, Cleland JGF, Coats AJS, Falk V, González-Juanatey JR, Harjola VP, Jankowska EA, Jessup M, Linde C, Nihoyannopoulos P, Parissis JT, Pieske B, Riley JP, Rosano GMC, Ruilope LM, Ruschitzka F, Rutten FH, van der Meer P; ESC Scientific Document Group. 2016 ESC Guidelines for the diagnosis and treatment of acute and chronic heart failure: The Task Force for the diagnosis and treatment of acute and chronic heart failure of the European Society of Cardiology (ESC)Developed with the special contribution of the Heart Failure Association (HFA) of the ESC. Eur Heart J. 2016 Jul 14;37(27):2129-2200. doi: 10.1093/eurheartj/ehw128. Epub 2016 May 20. Erratum in: Eur Heart J. 2016 Dec 30;: PMID: 27206819.

Ref 12: Riehle C, Bauersachs J. Small animal models of heart failure. Cardiovasc Res. 2019 Nov 1;115(13):1838-1849. doi: 10.1093/cvr/cvz161. PMID: 31243437; PMCID: PMC6803815.

so on...

Response:

Thank you for your indication. We recheck the form of reference and corrected surname/first names in the revised manuscript.

Reviewer 2 Report

Thank you for revising the manuscript. It can now be accepted for publication.

Author Response

Reviewer 2

Comment:

Thank you for revising the manuscript. It can now be accepted for publication.

Response:

                 We appreciate all of your insightful comments. Thank you for taking the time and energy to help us improve the paper.
